# High dielectric barium titanate porous scaffold for efficient Li metal cycling in anode-free cells

Chao Wang[1], Ming Liu[1], Michel Thijs[2], Frans G. B. Ooms[1], Swapna Ganapathy [1] & Marnix Wagemaker [1✉]

Li metal batteries are being intensively investigated as a means to achieve higher energy density when compared with standard Li-ion batteries. However, the formation of dendritic and mossy Li metal microstructures at the negative electrode during stripping/plating cycles causes electrolyte decomposition and the formation of electronically disconnected Li metal particles. Here we investigate the use of a Cu current collector coated with a high dielectric $BaTiO_3$ porous scaffold to suppress the electrical field gradients that cause morphological inhomogeneities during Li metal stripping/plating. Applying operando solid-state nuclear magnetic resonance measurements, we demonstrate that the high dielectric $BaTiO_3$ porous scaffold promotes dense Li deposition, improves the average plating/stripping efficiency and extends the cycling life of the cell compared to both bare Cu and to a low dielectric scaffold material (i.e., $Al_2O_3$). We report electrochemical tests in full anode-free coin cells using a $LiNi_{0.8}Co_{0.1}Mn_{0.1}O_2$-based positive electrode and a $LiPF_6$-based electrolyte to demonstrate the cycling efficiency of the $BaTiO_3$-coated Cu electrode.

[1] Section Storage of Electrochemical Energy, Radiation Science and Technology, Faculty of Applied Sciences, Delft University of Technology, Delft, Netherlands. [2] Neutron & Positron Methods for Materials, Radiation Science and Technology, Faculty of Applied Sciences, Delft University of Technology, Delft, Netherlands. ✉email: m.wagemaker@tudelft.nl

The development of high energy density batteries beyond the current Li-ion battery technology is necessary to meet the increasing demand of various applications such as electric vehicles[1–5]. Lithium metal is considered to be the ultimate anode material because it possesses the highest theoretical specific capacity (3860 mA h g$^{-1}$, approximately ten times that of current graphite-based anodes), and a low redox potential that endows it with a high battery output voltage (−3.040 V vs standard hydrogen electrode)[6–10]. However, upon repeated battery charging and discharging, the plating and stripping of Li metal induces irreversible processes that lead to fast capacity decay which drastically limits the Li metal battery cycle life[6–10]. The formation of high surface area dendritic and mossy Li metal morphologies, in combination with the fierce reactivity of Li metal with conventional non-aqueous organic electrolytes, leads to an irreversible loss of active Li. This is associated with the formation of a solid electrolyte interphase (SEI) and the formation of electronically disconnected Li metal particles. The latter is often referred to as "dead" Li, which has lost contact with the current collector[11–14]. Additionally, dendritic structures may penetrate the separator/electrolyte and reach the cathode, causing an internal short-circuit. This may in turn induce rapid spontaneous discharge and consequential safety hazards[6–10]. The loss of active Li in lab-scale cells is often masked by the excess of Li metal present. In practical cells, however, the amount of excess Li needed to compensate for these losses should be minimized, so as to maximize the energy density[15]. In an ideal scenario, the Li metal anode is completely absent from the battery in its initial discharged state and all Li is stored in the positive electrode. During the initial charging, Li ions are extracted from the positive electrode and deposited as Li metal on the Cu current collector[15]. This so-called "anode-less" or "anode-free" design, has the added benefit of eliminating the need for Li metal handling during production.

Strategies that aim to suppress, prevent and block dendrite formation are being intensively investigated[6–10,16,17], typically guided by the current understanding of dendrite nucleation and growth. The space charge model of Chazalviel predicts that when the Li-ion concentration on the surface of the negative electrode

drops to zero, after the characteristic Sand's time, plating becomes inhomogeneous and amplified growth of dendrites is induced[18]. These features necessitate the adoption of strategies that enhance ion mobility, increase the transference number and introduce a large negative electrode surface area. These strategies aim to promote a homogenous Li-ion flux and prevent ion depletion at the negative electrode surface[6–10,16,17]. Modelling of the early stages of nucleation and growth under heterogeneous electrodeposition indicates that surface inhomogeneities, particle size and wettability of the negative electrode play a critical role in facilitating dendrite formation[19]. This implies that dendrite growth can be steered by controlling these parameters as experimentally demonstrated[20]. When compared with Mg metal, which does not favour dendrite formation, it is found that both the high surface diffusion barriers for Li and the low surface energy density promote dendrite formation. This effectively allows the formation of large surface area Li metal morphologies[21]. To overcome or circumvent these issues, various strategies aimed at controlling the Li metal/electrolyte interface and the SEI composition have been considered[22]. Another factor that drives Li metal growth is residual stress within the Li metal itself[23]. This has motivated the design of substrates which are capable of releasing this stress[10,23]. Finally, studies on the effects of mechanical forces on dendrite nucleation and growth have shown that electrolytes with a shear modulus at least 2 times greater than that of Li metal, can prevent dendrite growth[24]. This has given rise to research where mechanically strong separators, solid electrolytes and protective films have been investigated[6–10,16,17].

The use of three dimensional (3D) scaffolds is an interesting approach, as they provide the possibility to control the interface, the local interface environment and to a large extent the charge transport. They can also provide a route to mitigate electrode delamination due to their ability to accommodate large volume changes upon Li metal plating and stripping. In a 3D scaffold that is electronically conducting, the electric field is expected to be roughly uniform and the local current density is reduced, suppressing dendrite growth[25–30]. However, the 3D porous conductive matrix is equipotential, due to which Li metal can also be deposited on top of it, thus negating its targeted function to contain Li-metal growth within the 3D porous matrix[31].

Here we explore an alternative approach through the introduction of an electronically insulating 3D scaffold with a high dielectric constant (which we will refer to as a high dielectric material). Due to the polarizing power of the high dielectric material, an effective immobile surface (space) charge density $\rho_{charge}$ is established, opposing the applied field in the battery that scales with the dielectric constant ($\nabla \cdot D = \rho_{charge}$ where $D = \varepsilon_0 \varepsilon_r E$, $\varepsilon_0$ and $\varepsilon_r$ the vacuum and relative permittivity and $E$ the electrical field). As a consequence, the electrical field lines are drawn towards the high dielectric material (dictated by Gauss Law) which leads to a lowering of the divergence of the electrical field in the vicinity of the Li metal deposition. At the tip of a dendrite near a high dielectric material, this effectively leads to a decrease in the electrical field divergence, and thus to a lower local electrical field gradient. This is shown in Fig. 1, where the calculated electrical field gradient near a dendrite with, and without high dielectric volumes in its vicinity are compared under very simplified conditions. This is proposed to lower the driving force for the plating of Li-ions at the tip of a Li metal inhomogeneity near to the high dielectric material. Indeed more homogeneous deposits have recently been observed due to the presence of high dielectric materials[32]. Based on this simple principle, we prepared porous 3D barium titanate (henceforth denoted as BTO) scaffolds which have a high dielectric constant ($\varepsilon_r \approx 4000$). The impact of this high dielectric based 3D scaffold

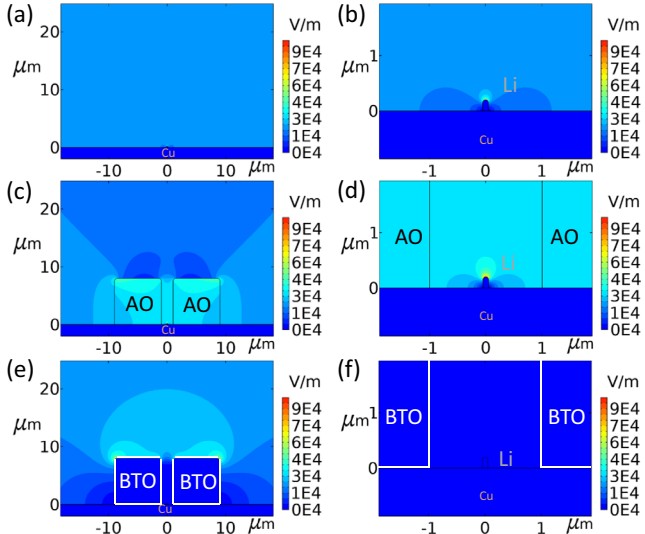

**Fig. 1 Electrical field simulations around a Li metal deposit with and without the presence of low and high dielectric blocks.** Li metal deposit **a** on bare Cu planar copper and **b** zoomed-in figure, **c** in combination with low dielectric AO block and **d** zoomed-in figure, **e** in combination with high dielectric BTO block and **f** zoomed-in figure.

is examined by direct comparison with an $Al_2O_3$ (henceforth denoted as AO) 3D scaffold having a low dielectric constant and a comparable 3D scaffold morphology. These Li metal-free scaffolds (BTO and AO) are deposited on a Cu substrate and cycled versus a Li metal electrode to evaluate their ability to obtain smooth and uniform Li deposits and to efficiently strip and plate lithium for a prolonged period of time. To highlight the impact of this approach, cycling has been performed in combination with a conventional electrolyte (1 M $LiPF_6$ in 1:1 v/v EC: DMC), presenting the worst-case scenario as this electrolyte is well known for leading to uncontrolled SEI formation. Bare Cu and the AO scaffold lead to the well established observations for this electrolyte i.e. a low Coulombic efficiency (henceforth denoted as CE), high overpotentials and a short cycle life. However, the presence of the insulating high dielectric porous BTO scaffold leads to a large increase in CE and small overpotentials. When used in a full cell configuration, the BTO scaffold paired with an NMC811 positive electrode, results in an improved CE and cycle life. Operando solid-state NMR measurements also performed on full cells indicate that the high dielectric scaffold induces more compact plating as compared to bare Cu, leaving no observable dead Li metal and resulting in less SEI species. From a practical perspective the porosity of the 3D BTO scaffold which is ~74%, leads to a relatively low specific capacity of ~800 mA h $g^{-1}$ due to the large weight of this scaffold. To increase this to 2000 mA h $g^{-1}$ for instance would only require 8% of the volume of the negative electrode to be BTO (a 92% porous scaffold). At present, the high dielectric constant of the scaffold is shown to suppress dendrite growth and promote homogeneous plating, therefore damping the self-amplifying cycle of SEI formation and dendritic growth. Nevertheless, several challenges remain, one of which is the reactivity of the BTO scaffold. Though very limited, it contributes to the initial capacity loss. The cycling of these anode-free scaffolds, still appears to terminate via a short circuit. This prompts the investigation of combinatorial approaches, especially utilizing optimized electrolytes and binders to improve performance. However, the proposed BTO scaffold is of practical interest because of the possibility to process the high dielectric material via an electrode casting method. In this fashion, the proposed strategy presents intriguing insights into the role of the dielectric constant on Li metal deposition. This provides an interesting research direction for the design of negative electrode substrates to achieve efficient Li metal stripping/plating in anode-free cells.

## Results and discussion

**Electrical field calculations**. To understand the impact of the dielectric constant of a dielectric block on the electrical field gradient at a nearby Li metal dendrite, the electrical field is simulated in two dimensions for simplified geometries. This is performed for both a bare Cu current collector with a Li metal dendrite, and a Cu current collector in combination with high dielectric BTO and low dielectric AO blocks surrounding the dendrite as shown in Fig. 1. The relative permittivity's of BTO, AO and the electrolyte (1 M $LiPF_6$ in 1:1 v/v EC: DMC) are 4000, 8 and 40, respectively.

These Cu, BTO/Cu and AO/Cu electrodes are placed against a counter electrode at a distance of 250 μm separated by an electrolyte to approximate the situation in a cell. The BTO and AO volumes are taken as 8 × 8 μm with a gap of 2 μm in which a lithium dendrite is placed, represented by a rectangular shape (0.1 × 0.2 μm) with a hemispherical tip. In Fig. 1a, b, the simulated electrical field around a single dendrite in the electrolyte is shown. Near the tip of the Li metal, the electrical field gradient increases, demonstrating that the electrical field

lines are focused at the tip of the electronically conducting Li metal dendrite. This is driven by the larger surface charge density present at the sharp electronically conducting features. This promotes the preferential deposition of Li-ions from the electrolyte on the tip of the dendrite, representing the fundamental driving force for dendrite formation in a homogeneous medium. As shown in Fig. 1c, d, volumes of AO (relative permittivity 8, smaller than that of the electrolyte) and in Fig. 1e, f, blocks of BTO (relative permittivity 4000, much higher than that of the electrolyte) are added on both sides of the Li -dendrite. For AO, the electric field gradient near the tip of the dendrite increases (Fig. 1d) when compared to its presence in the electrolyte alone (Fig. 1b), whereas for BTO, the electric field gradient at the tip of the dendrite disappears (Fig. 1e). The low polarizability of the AO "leaves" the electrical field gradient at the Li metal tip, whereas the high polarizability of the BTO "pulls" the electrical field lines away from the Li metal tip towards the surface of the BTO itself. This suggests that the presence of high dielectric volumes in the vicinity of a Li metal growth takes away the driving force for Li deposition at the tip of sharp features, thus taking away the driving force for tip driven dendrite growth. Several geometries were simulated, such as different volume distances, different dendrite lengths to name a few, provided in the Supplementary Information Supplementary Figs. 1–3, showing similar results. Clearly, these simulations represent a simplified condition that may overestimate the electric field gradient at the tip of a dendrite, as the electric field gradient will typically be dictated by the local environment. The simulations merely serve to make it plausible that in the vicinity of a high dielectric material, the electric field gradients are expected to reduce. Based on this, it can be postulated that a high dielectric scaffold can suppress dendrite growth and promote a homogeneous and more dense Li metal filling of the pores of the scaffold. This can be expected to lead to less "dead" Li metal forming as well as a smaller Li metal electrolyte interface, thus less electrolyte decomposition, enabling higher cycling efficiencies and a longer cycle life of the Li-containing negative electrode.

**Physicochemical characterization of the Cu-based electrodes**. To evaluate the impact of the BTO and AO porous scaffolds, both materials were cast on a Cu current collector, similar to what is done in the preparation of regular Li-ion insertion electrodes. Supplementary Fig. 4 shows the morphology of the BTO and AO particles and the as-prepared electrodes obtained via scanning electron microscopy (SEM) measurements. Both materials have an average particle size of approximately 8 μm as seen from the SEM images (Supplementary Fig. 4 a, c) and via a Dynamic Light Scattering (Supplementary Fig. 5) analysis. According to the nitrogen adsorption/desorption isotherms (Supplementary Fig. 6), their Brunauere–Emmette–Teller (BET) specific surface areas are also comparable i.e. 23.59 and 23.08 $m^2 g^{-1}$ for BTO and AO, respectively. The comparable surface area and particle size distribution of BTO and AO are crucial for enabling us to seperate the impact of the dielectric constant on Li metal deposition from that of the scaffold morphology. These ball-milled materials were used to build the scaffold for lithium metal plating on a Cu current collector with the $NH_4HCO_3$ template to achieve a high porosity[33]. This is required to achieve high specific negative electrode capacities, taking into account the weight of the scaffold. The resulting porosity of the AO and BTO scaffolds are 68% and 74%, respectively and are thus comparable. Assuming that this porous volume is completely filled by Li metal, the specific capacity is decreased due to the weight of the scaffold, which for BTO ($\rho = 6.02$ g $cm^{-3}$) results in a specific capacity of approximately 800 mA h $g^{-1}$. To achieve a specific capacity of 2000 mA h $g^{-1}$ demands approximately only 8% of

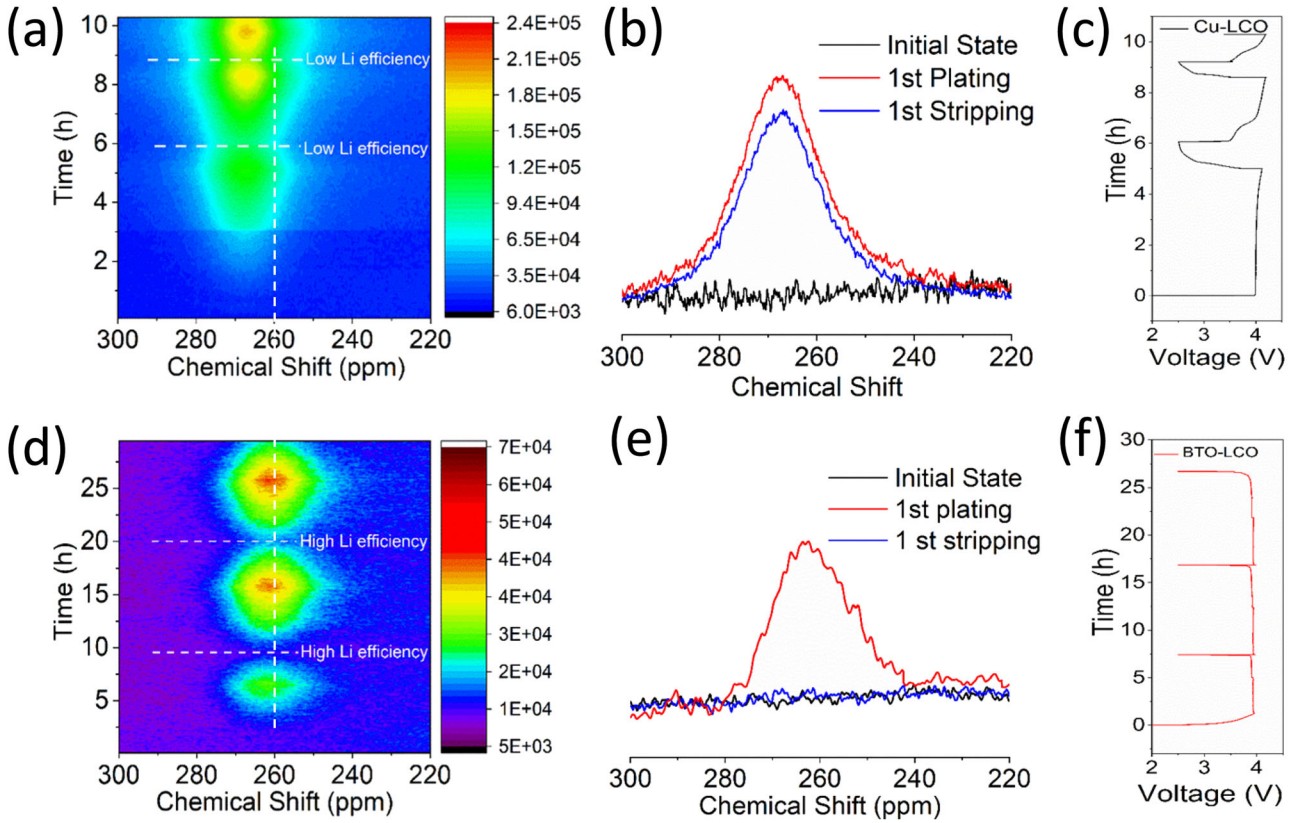

**Fig. 2 Operando $^7$Li solid-state NMR of Li metal plating/stripping on bare Cu and the BTO-coated Cu substrates in full cell configuration. a–c** Bare Cu versus a LiCoO$_2$ positive electrode with a 1 M LiPF$_6$ EC/DMC electrolyte cycled at 0.2 mA cm$^{-2}$ to 1 mA h cm$^{-2}$ charge capacity and discharged to 2.5 V cut-off. **a** 2D $^7$Li operando NMR spectra as a function of cycling, **b** 1D $^7$Li solid-state NMR spectra at selected conditions, **c** voltage profile. **d–f** Anode-free BTO scaffold on Cu versus a LiCoO$_2$ positive electrode with a 1 M LiPF$_6$ EC/DMC electrolyte cycled at 0.2 mA cm$^{-2}$ to 1 mA h cm$^{-2}$ charge capacity and discharge to 2.5 V cut-off. **d** 2D $^7$Li operando NMR spectra as a function of cycling, **e** 1D $^7$Li solid-state NMR spectra at selected conditions, **f** voltage profile.

BTO volume (a 92% porous scaffold). To elucidate the impact of a high dielectric scaffold on the cycle life of the anode-free configuration of these electrodes, a blank 1 M LiPF$_6$ in EC/DMC electrolyte was used. As mentioned earlier, this represents a worst-case scenario because it is well known that this electrolyte formulation leads to uncontrolled dendrite and SEI growth[9,34].

**Operando solid-state NMR characterization.** To test the hypothesis that the Li metal deposition is less dendritic and more homogeneous in the pores of the high dielectric scaffold, and that this leads to better reversibility, operando $^7$Li solid-state NMR has been performed, the results of which are shown in Fig. 2. Operando $^7$Li solid-state NMR is a direct probe of Li in realistic cell conditions and allows us to distinguish differences in the Li metal microstructure[12,35,36]. As shown in Supplementary Fig. 7, the signals from the Li-species in the SEI and the electrolyte are in the expected region for diamagnetic materials (-10 to 10 ppm). Li in the LiCoO$_2$ (positive electrode) is seen in the wide region of −50 to 50 ppm. Both these resonance signatures can readily be separated from the Li metal chemical shift (~245–270 ppm) which is dominated by its Knight shift[35]. The NMR radio-frequency (RF) has a limited penetration depth into the Li metal, referred to as skin depth, which at the presently employed $\mathbf{B_0}$ field of 11.7 T is approximately 11 μm[13]. As a consequence, for deposited Li-microstructures such as dendrites and mossy structures, which are typically smaller than a few micrometres, complete RF penetration is expected. A Li metal strip[13,35] gives rise to a resonance signal at ~245 ppm when placed perpendicular

to the fixed magnetic field $\mathbf{B_0}$ and at ~270 ppm when the strip is parallel to $\mathbf{B_0}$, a consequence of the bulk magnetic susceptibility effect[13]. An anode-free design (absence of Li metal at the negative electrode in the initial state) is investigated by operando solid-state NMR, utilizing LiCoO$_2$ as the counter electrode (and Li source), such that the observed Li metal signal arises only due to the deposition at the negative electrode of interest, similar to a recently reported study[37].

During charge (plating) on the bare Cu current collector, the $^7$Li metal resonance appears at approximately 266 ppm as shown in Fig. 2a, b. On the other hand for the Li metal deposits in the BTO scaffold, it occurs at approximately 260 ppm as shown in Fig. 2d, e. Dendritic microstructures, growing perpendicular to Cu have been associated with a narrow chemical shift range centred at around 270 ppm whereas mossy microstructures encompass broader spectral region covering a chemical shift range of 262–274 ppm[12]. Not surprisingly, the present results demonstrate that on a bare Cu current collector, in combination with a conventional carbonate-based electrolyte, mossy/dendritic Li metal growth is already initiated during the first plating cycle. Interestingly, the Li metal chemical shift in the BTO scaffold is similar to that observed for dense Li metal plated in the separator when plated under pressure[12]. The density of the BTO scaffold is similar to that of the separators, thus the operando $^7$Li solid-state NMR indicates that on Li metal plating, the BTO scaffold pores are filled, suppressing the growth of mossy and dendritic microstructures. Discharge (Li metal stripping) of the bare Cu current collector results in high overpotentials, indicating a highly

increased internal resistance, which is associated with severe SEI formation that hinders Li-ion transport as seen in Fig. 2c. The intensity of the [7]Li metal NMR resonance only slightly decreases during discharge (Fig. 2a, b), reflecting the difficulty in stripping the Li metal deposits from the Cu surface. On cycling this results in rapid accumulation of inactive "dead" Li metal (see Fig. 2a) comprehensively studied recently using operando NMR[37]. In contrast, the voltage during discharge (Li metal stripping) for the BTO scaffold is nearly the same as during charge (see Fig. 2f), indicating that the internal resistance is practically unaffected by the SEI formation. Concurrently, the [7]Li metal resonance completely disappears upon the charge, as confirmed in Fig. 2e, demonstrating that all Li metal can be completely stripped from the BTO scaffold, and thus that a high Li metal stripping/plating efficiency is achieved. The large increase in the Li metal signal after the first cycle can be explained by a low first cycle efficiency, where a significant fraction of the capacity is lost to the formation of SEI products by introducing Li metal at the negative electrode. In addition, the spectral region around the [7]Li chemical shift of 0 ppm is also compared, where the Li resonance of the diamagnetic SEI and electrolyte appear as seen from Supplementary Fig. 8. We are unable to resolve the SEI species from the electrolyte; however, we do observe an increase in the signal over the first three cycles for the bare Cu current collector, which is not observed for the BTO scaffold. This suggests that much less electrolyte decomposition occurs for the latter, consistent with the more compact plating observed. In summary, operando [7]Li solid-state NMR demonstrates that the presence of the high dielectric anode-free BTO scaffold results in more compact and less dendritic/mossy Li metal, in line with the suggested impact of a high dielectric scaffold on the electric field gradients shown in Fig. 1, which reduces the amount of observed "dead" Li metal and SEI formation, even in a conventional 1 M LiPF$_6$ EC/DMC electrolyte.

**Electrochemical performance of Cu-based electrodes.** Galvanostatic cycling in the half cell is carried out, using a Li metal foil as the counter electrode, to compare the reversibility of anode-free bare Cu with the porous BTO and AO scaffolds. Different cycling conditions are investigated: (1) constant capacity plating followed by constant capacity stripping (if the plated capacity is not reached, a 1 V vs Li/Li$^+$ cutoff voltage is employed, (2) constant capacity plating, followed by a 1 V vs Li/Li$^+$ cutoff voltage on stripping. The resulting CE values (ratio of the Li stripping capacity to the plating capacity) are shown in Fig. 3 and a selection of the corresponding voltage profiles in Fig. 4.

The CE in the half cell configuration, with bare Cu, AO and BTO working electrodes allows the quantification of Li loss upon stripping. This can have two sources, namely "dead" Li (Li metal electrically isolated from the current collector) and SEI species that are formed due to electrolyte reduction[38]. On comparing the BTO scaffold (high dielectric) with the AO scaffold (low dielectric), it is seen that the BTO-based scaffold maintains a much higher CE over several additional cycles as shown in Fig. 3. However, this also depends on the cycling protocol, as is demonstrated by comparing the two half cell tests protocols i.e. (1) and (2) as shown in Fig. 3a, b, respectively. In both cases, the first two cycles are performed at a relatively low current density of 0.5 mA cm$^{-2}$ (to form the SEI), and during these cycles, the stripping is terminated when the voltage cutoff is reached (1 V vs Li/Li$^+$) resulting in low CE values i.e. in the order of ~60% and ~85%. For the BTO cycling protocol (1) this has interesting consequences. After the two initial cycles, the stripped capacity reaches the plated capacity (before the voltage cutoff is reached), and thus the cycling is restricted by the capacity (reaching the

plated capacity), resulting in a CE of 100% that is maintained for over 240 cycles before it suddenly drops. Clearly, this 100% CE does not imply that there is no Li loss in the form of dead Li or SEI species during these cycles, especially since this is a conventional electrolyte solution known to result in rapid capacity loss as observed for the bare Cu and AO scaffold. The CE that quantifies the Li loss at the BTO-coated Cu electrode is the average CE over all cycles, including the first two cycles, which amounts to 99.82% (1 mAh cm$^2$ at 2 mA cm$^{-2}$) for BTO. This is a large improvement in comparison with the bare Cu and AO scaffold, and remarkable given the non-optimized electrolyte used. Similar results are also obtained for BTO at higher current densities shown in Fig. 3c, d (current densities of 4 mA cm$^{-2}$ and 8 mA cm$^{-2}$ to a 1 mA h cm$^{-2}$ plated capacity results in average CE values of 99.35% and 99.30%, respectively). When cycling towards a larger (practical) capacity of 4 mA h cm$^{-2}$ at a current density of 4 mA cm$^{-2}$ this results in an average CE of 99.68% that can be maintained for over 160 cycles (Supplementary Fig. 9). A more common cycling strategy is to terminate the Li metal stripping by the cutoff potential alone, in this case at 1 V vs Li/Li$^+$, the results of which are shown in Fig. 3b i.e. when cycling protocol (2) is utilized. This cycling protocol results in a constantly low CE for AO, and a strongly fluctuating CE for BTO maintaining values at around 100% over several cycles. From the voltage profiles displayed in Supplementary Fig. 10, we can confirm that no short-circuits occur in the cells and speculate that the fluctuating efficiency seen in Fig. 3b is associated with the build-up of the Li metal reservoir during the initial stripping/plating cycles. We suggest the following mechanism to explain the CE values observed for both half cell cycling strategies. As expected, the CE during the initial cycles is similar for both cycling protocols as seen in Fig. 3a, b; however, in subsequent cycles, the CE achieves values above 100% typically followed by CE's below 100% and again by values above 100% etc. This suggests that during the initial cycles, a reserve of dead Li metal is built up (CE < 100%), that can be utilized in subsequent cycles (CE > 100%) because it is reconnected during plating. This again depletes the Li metal reserve, leading to a plating/stripping cycle with a lower CE, again building up the dead Li metal reserve. This can only be sustained if the fraction of capacity loss towards irreversible SEI formation is small, which appears to be the case for BTO and not for bare Cu and AO. For capacity restricted cycling in Fig. 3a, this also explains why a CE of 100% can be maintained for many cycles by the BTO scaffold electrodes. The Li metal reservoir formed during the initial cycles effectively leads to the transition from a BTO‖Li cell to a BTO-Li‖Li cell, which can be sustained until the reserve runs out, presumably by the ongoing SEI formation and formation of dead Li metal that cannot be reconnected. As suggested above, in this case the only meaningful CE in this situation is the average CE, including the initial cycles during which the reservoir is built up. Regardless of the impact of the cycling strategy, the results indicate that for BTO, the dead Li that forms can more easily be reconnected, in line with more compact plating deduced from the operando [7]Li NMR experiments (Fig. 2), which leads to less irreversible Li loss to SEI species, compared to the AO and bare Cu negative electrodes.

The evolution of the voltage during the anode-free half cell cycling protocol (1) and (2), at a 2 mA cm$^{-2}$ current density to a 1 mAh cm$^{-2}$ capacity are shown in Fig. 4a, b, and in Fig. 4c, d for current densities of 4 and 8 mA cm$^{-2}$. In addition, detailed voltage profiles are provided in Supplementary Figs. 10 and 11. Under all cycling conditions tested, the anode-free Cu and AO scaffold result in a rapid increase in the plating and stripping potentials. This can be associated with the SEI formation, amplified by mossy/dendritic

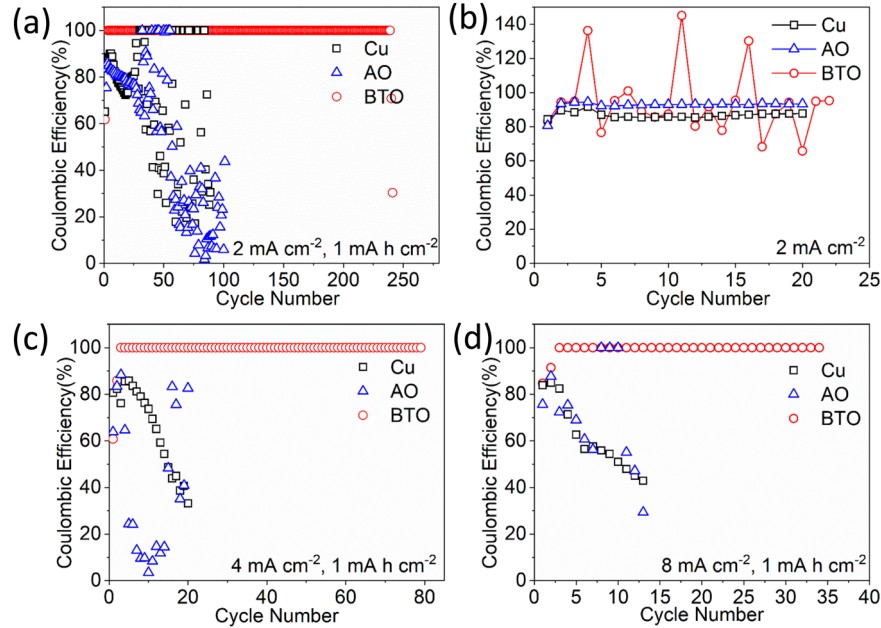

**Fig. 3 Half-cell performance of the anode-free Cu and AO and BTO scaffolds versus a Li metal electrode.** Lithium Coulombic efficiency **a** with a fixed areal plated capacity of $1\,mA\,h\,cm^{-2}$ and stripping terminated by reaching the plated capacity or a 1 V vs Li/Li$^+$ voltage cutoff **b** with stripping terminated by a 1.0 V vs Li/Li$^+$ voltage cutoff at $2\,mA\,cm^{-2}$. Coulombic efficiency for cells performed at the current density of **c** $4\,mA\,cm^{-2}$ and **d** $8\,mA\,cm^{-2}$ with the anode-free half-cell cycling protocol (1).

Li metal growth, leading to a high internal resistance until no cycling capacity remains, or a short-circuit occurs. The BTO scaffold can maintain much lower overpotentials for more cycles, i.e. approximately 68, 220 and 375 mV at current densities of 2, 4 and 8 $mA\,cm^{-2}$, respectively. Nevertheless, even for the BTO scaffold, the overpotentials increase indicating that upon extended cycling a number of the cells fail by a short-circuit, reflecting severe SEI growth. The half cells with a BTO scaffold cycling at current densities of 2 $mA\,cm^{-2}$ and 4 $mA\,cm^{-2}$ fail by a short-circuit at cycle number 241 and 80, respectively. The cell cycling at 8 $mA\,cm^{-2}$ fails due to the high overpotential after 30 cycles and the cell cycled to a higher capacity (4 $mA\,h\,cm^{-2}$ at 4 $mA\,cm^{-2}$) fails by a short-circuit at the 167$^{th}$ cycle. These observations are further supported by electrochemical impedance spectroscopy (EIS) measurements shown in Fig. 4e–g. The total resistance of the cells with the Cu and AO scaffold negative electrode shows a strong increase after 10 cycles at a 2 $mA\,cm^{-2}$ current density as shown in Fig. 4g, associated with the SEI formation driven by mossy/dendritic Li metal growth. Under the same conditions, the BTO scaffold has a much lower interfacial resistance, indicating less SEI formation.

Full cell tests are performed, comparing Li metal foil with precycled (anode-free) BTO negative electrodes, each in combination with an NMC811 electrode (~5 $mg\,cm^{-2}$) employing cutoff voltages of 3 V during discharge and 4.3 V vs Li/Li$^+$ during charge. Also in this case, a conventional 1 M LiPF$_6$ EC/DMC electrolyte is employed. The resulting CE values are provided in Fig. 5 and voltage profiles in Fig. 6. The BTO scaffold outperforms the Li metal negative electrode in cycling rate and stability as shown in Fig. 5. The coulombic efficiency of the full cell is on average 99.37% over ~70 cycles with a prelithiated BTO electrode. Evolution of the voltage of BTO scaffold when paired with an NCM positive electrode at 0.33 $mA\,cm^{-2}$ at different cycles also manifested lower overpotentials than Li metal (Fig. 6). Nevertheless, the cycling is less efficient when compared to the half cell, which in addition to the formation of dead Li and SEI species at the negative electrode may be due to the crossover of transition metal ions from the NMC811 positive electrode, and

other sources of irreversible capacity loss at the positive electrode side (e.g. reactivity with the electrolyte, loss of active material).

With respect to the irreversible capacity loss, it is important to establish the specific contribution from BTO. As shown in the cyclic voltammetry curves in Supplementary Fig. 12, the BTO scaffold shows a small peak at 1.2 V (and no activity is observed for AO). However, there is no structural change of the BTO scaffold after the 1$^{st}$ and 100$^{th}$ cycles as verified by XRD (Supplementary Fig. 13). Therefore the BTO scaffold does contribute to the initial capacity loss. It does not however show large continuing degradation when in contact with Li metal, and upon extended cycling. Although both BTO and AO can be reduced by Li metal, it appears that this process is passivated, most likely only occurring at the BTO/AO surface.

**Ex situ morphological investigations of the Li-hosting electrodes.** SEM measurements are carried out before and after cycling the cells at a current density of 2 $mA\,cm^{-2}$ to a specific capacity of 1 $mA\,h\,cm^{-2}$. This is done to compare bare Cu, the AO scaffold and the BTO scaffold, where the top images and the cross sections are shown in Fig. 7. The BTO scaffold appears to lead to more uniform lithium deposits suggesting that the deposited Li metal is more densely confined in the 3D scaffold of the BTO layer consistent with the operando NMR measurements depicted in Fig. 2. In contrast, Li deposition on the bare Cu foil exhibits islands of accumulated dendritic structures. Also dendrites are observed at the surface of the AO scaffold.

## Discussion

As schematically shown in Fig. 8a, a fundamental driving force for the formation of dendritic/mossy Li metal are large gradients in the electric field lines locally at the surface of the Li metal caused by surface inhomogeneities. This acts as a starting point for uncontrolled SEI formation which in turn amplifies further inhomogeneous mossy/dendritic Li metal deposition and the accumulation of inactive Li metal, as demonstrated by the

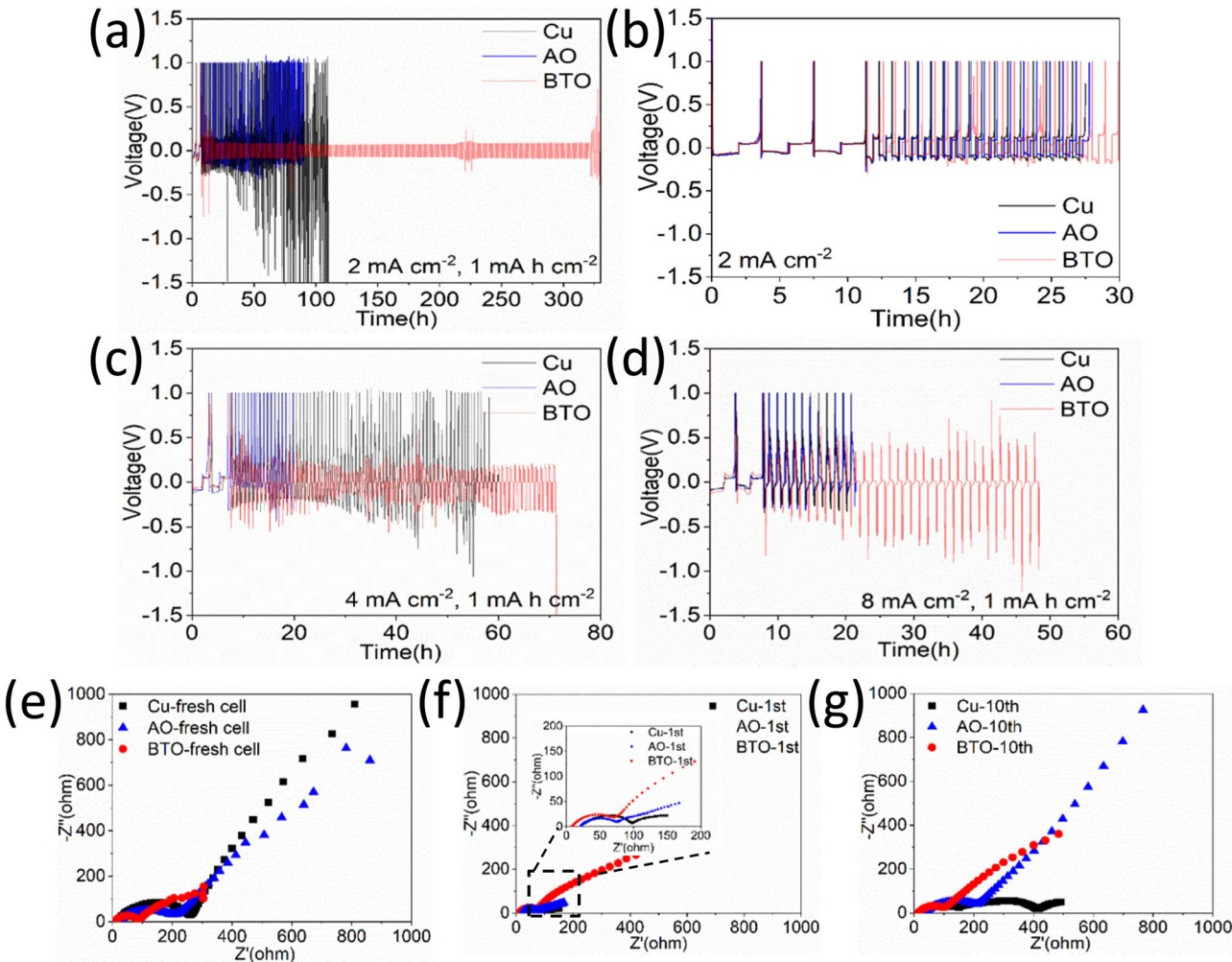

**Fig. 4 Half-cell voltage evolution of the anode-free Cu and AO and BTO scaffolds vs a Li metal electrode.** Evolution of the voltage during Li plating/stripping **a** with a fixed areal capacity of 1 mA h cm$^{-2}$ and **b** with a voltage limit at 1.0 V vs Li/Li$^+$ at 2 mA cm$^{-2}$ using three different electrodes. Evolution of the voltage of cells performed at the current density of **c** 4 mA cm$^{-2}$ and **d** 8 mA cm$^{-2}$ with the anode-free half-cell cycling protocol (1). Electrochemical impedance spectra of the half-cells for the bare Cu and the AO and BTO scaffolds **e** fresh cells, **f** the 1$^{st}$ cycle and **g** the 10$^{th}$ cycle measured at a current density of 2 mA cm$^{-2}$.

operando solid-state NMR (Fig. 2) and the SEM measurements (Fig. 7). This drives up the internal resistance (Fig. 4), eventually leading to cell failure. A comparison of the high dielectric BTO scaffold ($\varepsilon_r \approx 4000$) and the low dielectric AO scaffold ($\varepsilon_r \approx 8$), having a similar morphology (in terms of loading, particle size distribution and porosity), makes it possible to distinguish between the impact of a non-conducting scaffold morphology and the impact of the dielectric constant of the scaffold on the Li metal plating. As schematically shown in Fig. 8b, the 3D high-dielectric BTO scaffold is suggested to promote a decrease in the electric field gradients at the tip of uneven Li metal deposits, as supported by static electric field calculations (Fig. 1). These simulations are performed under highly simplified conditions, not taking into account the local polarization effects in the double layer of the Li metal. Nevertheless, it serves as an indication that high dielectric volumes can lower the electric field gradients at the tip of irregular Li metal morphologies between these volumes. It should also be noted that porous metal scaffolds are anticipated to have the same impact, as the dielectric constant of a metal is infinite by definition. The difference is however that an electronically conducting scaffold will allow dendrite growth on top of the scaffold, whereas the current high dielectric scaffold is electronically insulating, promoting plating inside the scaffold.

Lowering the electric field gradient inside the high dielectric BTO scaffold leads to a more homogeneous Li-ion flux, and thus to less mossy/dendritic and more homogeneous Li metal deposits as recently shown by Guo[32] and as evidenced by operando solid-state NMR (Fig. 2). This prevents the accumulation of deactivated Li metal, or if deactivated Li metal forms it is easily reconnected upon plating due to the more compact nature of the plated morphology, which is held responsible for the highly improved cycling efficiency (Fig. 3). The more compact nature of the deposits can be argued to lower the exposed Li metal/electrolyte contact area, diminishing SEI formation and thus delaying the uncontrolled and self-amplified growth of the SEI and dendritic Li metal deposits, rationalizing the low overpotentials during extended cycling as observed in Fig. 4. Notably, these results are achieved under the challenging conditions of an anode-free design i.e. the initial absence of Li metal in the BTO scaffold, and in combination with a non-optimized carbonate electrolyte (1 M LiPF$_6$ EC/DMC). The BTO scaffold appears to contribute to the poor CE during the first two cycles, as reflected in the capacity shoulder above 1 V vs Li/Li$^+$ in Fig. 4 also observed during the CV cycling (Supplementary Fig. 12). Lithiation of BTO can be expected to reduce Ti$^{4+}$, which will lower the dielectric constant, and thus lower the modulation of the electrical field. We suggest

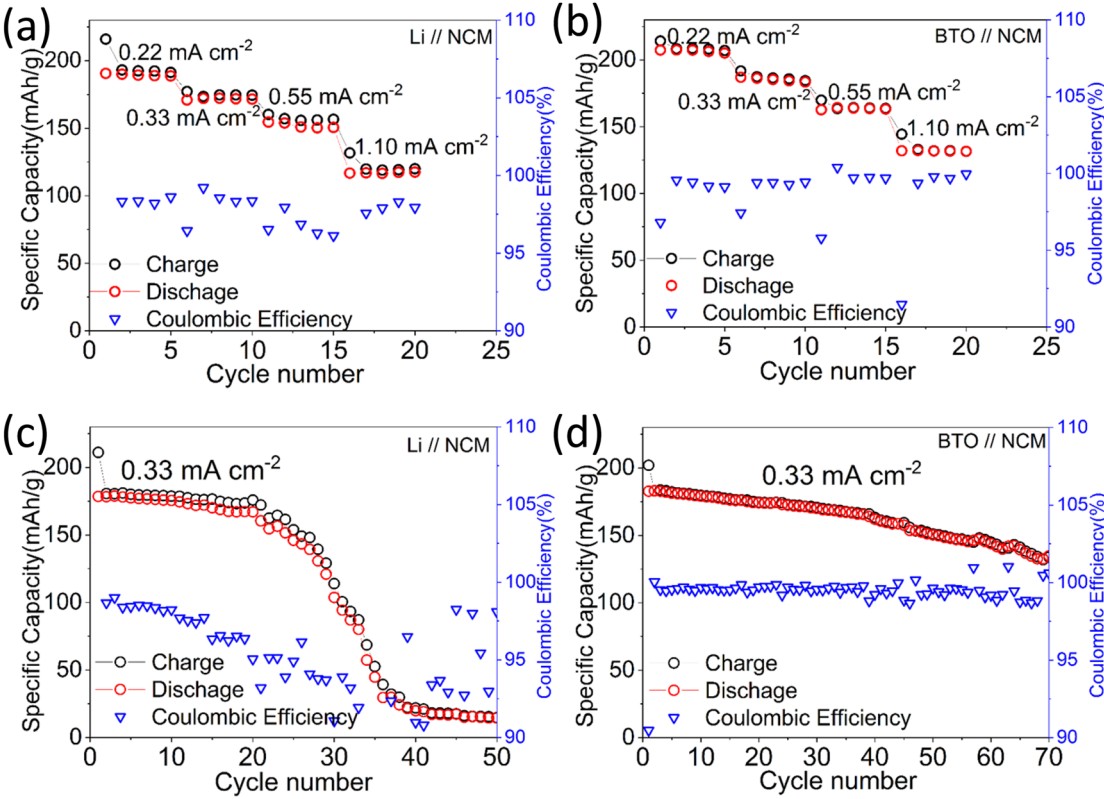

**Fig. 5 Electrochemical performances and coulombic efficiency of Li metal and BTO scaffold when paired with NCM positive electrode. a, b** Rate test from 0.22 mA cm$^{-2}$ to 1.10 mA cm$^{-2}$. **c, d** Cycling performance at 0.33 mA cm$^{-2}$.

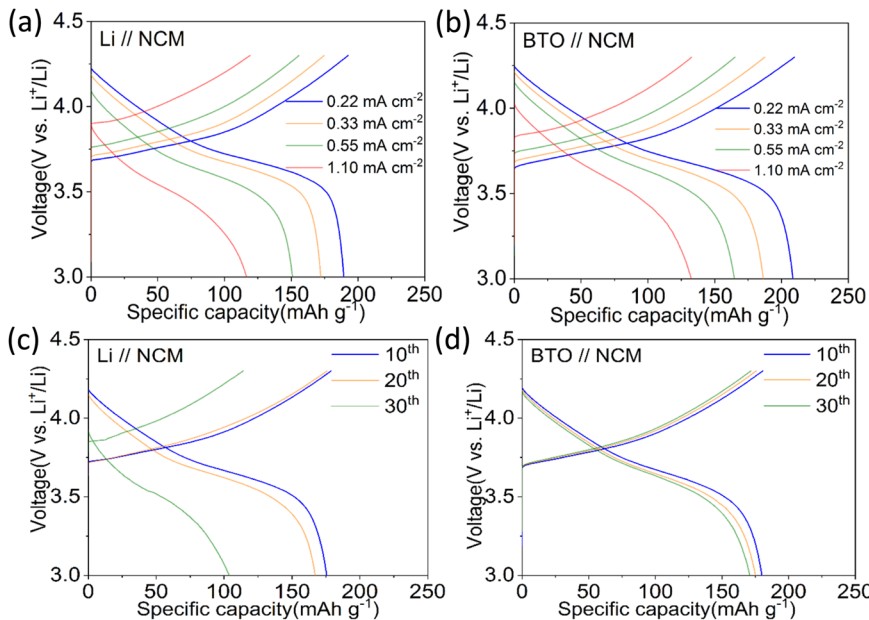

**Fig. 6 Voltage profiles of Li metal and BTO scaffold when paired with NCM positive electrode. a, b** Rate test from 0.22 mA cm$^{-2}$ to 1.10 mA cm$^{-2}$. **c, d** Voltage profiles when cycled at 0.33 mA cm$^{-2}$ at different cycle numbers.

that due to its poor electronic conductivity, BTO lithiation will be limited content with XRD analysis after cycling where structural changes are not seen (Supplementary Fig. 13). Another aspect that needs to be considered is the piezoelectric property of BTO, which implies that a local electric field can induce strain in BTO thus changing the porosity of the scaffold, which may impact dendrite growth. Considering that the electric field in the

simulations does not exceed 10$^5$ V/m, which is likely an over-estimation, the strain is expected to be well below 1%[39] and thus will cause a negligible change in porosity of approximately 70% porous scaffolds.

In conclusion, the challenge for Li metal electrodes is in preventing dendritic and mossy Li metal growth that catalyzes electrolyte decomposition and leads to low electrochemical

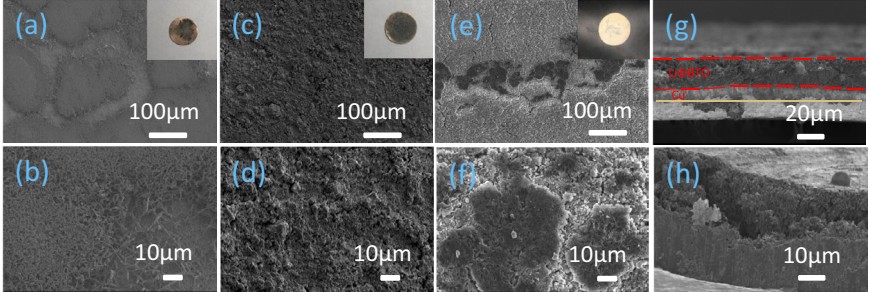

**Fig. 7 Ex situ SEM postmortem morphological investigations of the bare Cu, AO and BTO scaffold after plating to 1 mA h cm$^{-2}$ at a current density of 2 mA cm$^{-2}$. a, b** Bare copper electrode and zoomed-in figure, **c, d** BTO scaffold and zoomed-in figure, **e, f** AO scaffold and zoomed-in figure, **g, h** cross-section of BTO and Cu electrode after depositing 1 mA h cm$^{-2}$ Li metal at a current density of 2 mA cm$^{-2}$. Insets show the digital images of the complete electrodes.

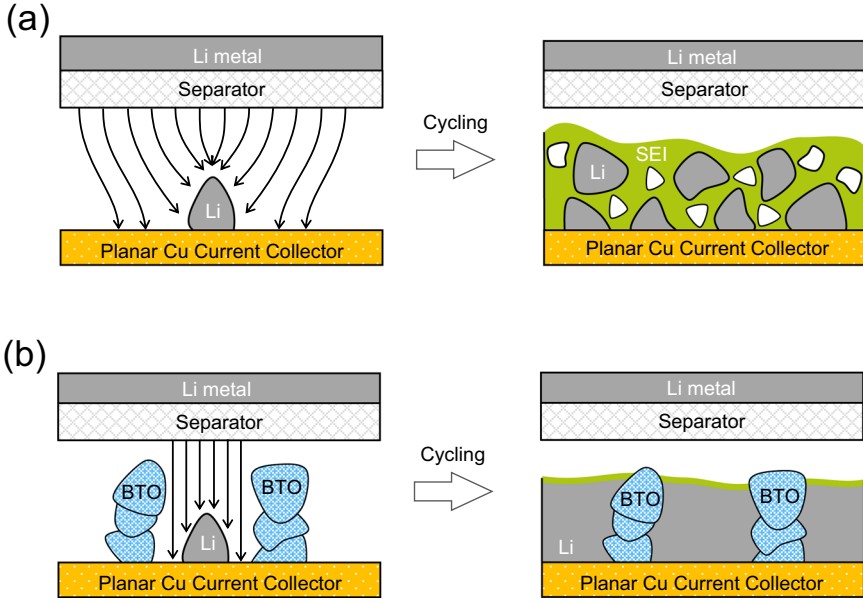

**Fig. 8 Schematic diagram of lithium metal plating and stripping with and without the presence of the high dielectric BTO porous scaffold. a** Lithium metal plating/stripping process on planer copper, **b** lithium metal plating/stripping process on BTO-coated copper foil.

plating/stripping efficiencies. Simplified static electrical field calculations suggest that the presence of a high dielectric material reduces the electric field gradient at the tip of a Li metal deposit that is micrometers away from the dielectric volume, suggesting that the Li metal dendritic and mossy microstructure growth can be suppressed by a negative electrode comprising of a high dielectric porous scaffold. 3D porous scaffolds are prepared through a facile casting approach using both a high dielectric scaffold material, BaTiO$_3$ (BTO) and a low dielectric scaffold material, Al$_2$O$_3$ (AO), to distinguish between the impact of a porous scaffold and the high dielectric constant on the electrochemical Li metal plating. Although these electronically insulating scaffolds show minimal electrochemical activity, the associated capacity loss does pose a challenge that needs to be addressed. The added weight due to the scaffold material lowers the specific capacity of the Li metal electrode, to approximately 800 mA h g$^{-1}$, and to achieve higher practical capacities this requires a lower volume of BTO in the negative electrode. Operando $^7$Li solid-state NMR of a BTO-coated Cu||LiCoO$_2$ cell demonstrates that the high dielectric scaffold induces compact plating and efficient stripping leaving practically no dead lithium behind after stripping. Half-cells with a BTO scaffold cycled against Li metal exhibit an average CE of 99.82%, low overpotentials and an extended cycle

life, in combination with a 1 M LiPF$_6$ EC/DMC electrolyte which can be considered as a worst-case scenario. With the same basic electrolyte, full cells also demonstrate improved performance with an average coulombic efficiency of 99.37%. These results suggest that high dielectric scaffolds provide an interesting strategy to improve the reversibility and safety of Li metal electrodes in an anode-free configuration. The next avenues to explore are combinations with more stable SEI forming electrolytes and additives and optimization of the high dielectric scaffold, to minimize the capacity loss during the first cycles and to further extend the cycle life.

## Methods

**Preparation of electrodes and electrochemical tests**. Commercial BaTiO$_3$ (Euro Support Advanced Materials B.V., denoted as BTO) and Al$_2$O$_3$ (Sigma-Aldrich, denoted as AO) powder were used to prepare 3D scaffolds on Cu (10 μm, non-dendritic, 99.99%) current collectors having a high relative permittivity ($\varepsilon_r \approx 4000$) and a low relative permittivity ($\varepsilon_r \approx 8$), respectively. Firstly, BTO and AO were ball-milled at 450 rpm, 6 h under Ar atmosphere using 10 ZrO$_2$ balls to achieve relatively small particles and similar particle size distribution. Both materials were mixed with polyvinylidene fluoride(PVDF) and NH$_4$HCO$_3$ (ratio at 5:1:4) using N-methyl-2-pyrrolidone (NMP) solvent to obtain a slurry which is casted on copper. The NH$_4$HCO$_3$ acts as a template to achieve a high porosity[33] to increase the specific capacity of the negative electrode (BTO scaffold). The porosity of the reported electrodes is 74% based on the loading and thickness. The resulting specific capacity, taking into account the weight of the BTO scaffold, equals (2046/

(0.53 + ((1/$p$)−1))) using the density of BTO (6.02 g/cm$^3$), Li metal (0.53 g/cm$^3$) and the specific capacity of Li metal (3860 mAh/g) and where $p$ represents the fractional porosity (between 0 and 1).

After that, the electrodes were dried under vacuum at 80 °C to remove the NH$_4$HCO$_3$ template. The resulting electrodes were cut into round electrodes with a diameter of 12.7 mm. Coin cells (CR2032) were assembled using as-prepared electrodes with lithium metal (11 × 0.6 mm, 99%) as a counter electrode, a polyethylene (PE) (Celgard 2300) separator and 150 µl carbonate electrolyte (1 M LiPF$_6$ in 1:1 v/v EC: DMC, water 10 ppm). Galvanostatic cycling was performed by deposition of Li onto the bare Cu working electrode or AO/BTO scaffolds coated on Cu with different current densities (2 mA cm$^{−2}$ to 8 mA cm$^{−2}$) to a fixed capacity (1 mA h cm$^{−2}$ or 4 mA h cm$^{−2}$), followed by Li stripping at different current densities up to a capacity limited method or a voltage cutoff of 1.0 V vs Li/Li$^+$. A rest time of 2–30 min was set between plating and stripping. Full cells composed of a porous BTO scaffold on a Cu current collector (denoted as BTO) in combination with a LiNi$_{0.8}$Co$_{0.1}$Mn$_{0.1}$O$_2$ (Umicore N.V., denoted as NCM) positive electrode were assembled. The BTO scaffold negative electrode was first cycled versus Li metal for two cycles to minimize initial irreversible Li consumption. NCM and LiCoO$_2$ (Sigma-Aldrich, denoted as LCO) positive electrodes were prepared by mixing the active material with Super P and PVDF in a mass ratio of 8:1:1, and NMP was used as a solvent. The mass loading of the NCM and LCO electrodes was ~5 mg cm$^{−2}$. Li/NCM and BTO/NCM cells were cycled within the potential range of 3.0–4.2 V (vs. Li/Li$^+$) at 25 °C. Galvanostatic cycling was conducted on a Maccor battery testing system. The impedance measurements of the coin cells were carried out on Autolab between 100 kHz and 0.01 Hz. Cyclic voltammograms (CVs) were recorded using the same electrochemical workstation at a scan rate of 1 mV s$^{−1}$ in the range of −0.5 V–3 V.

**Characterization of the materials and the electrodes**. SEM images were obtained of the 3D scaffolds after a discharge capacity of 1 mA h cm$^{−2}$. Before SEM imaging, the electrodes were rinsed with dimethyl carbonate in a glove box to remove lithium salts and dried several times in a vacuum chamber. Cross-section SEM samples were prepared by cutting with scissors in the glove box. Subsequently, samples were transferred into an SEM (JEOL JSM-6010LA) machine under dry Argon conditions with a closed box, and images were taken using an accelerating voltage of 2–10 kV (secondary electron). Nitrogen adsorption–desorption isotherms were recorded using an automatic surface area and porosity analyzer (Micromeritics) at 77 K. The particle size distribution of BTO and AO after ball milling was measured using Microtrac S3500.

LCO/Cu and LCO/BTOCu cells in a plastic cell capsule suitable for operando NMR measurements with non-aqueous carbonate electrolyte solutions were assembled in the glove box and studied by operando $^7$Li-NMR to monitor the microstructural evolution of Li deposits. Measurements were done on a wide bore Bruker Ascend 500 system equipped with a NEO console with a magnetic field strength of 11.7 T and a $^7$Li resonance frequency of 194.37 MHz. Operando static $^7$Li NMR experiments were performed at room temperature with an NMR Service ATMC operando NMR probe, and the electrochemical cell was simultaneously controlled by a portable Maccor battery testing system. During the 1D static $^7$Li NMR measurements, the cells were charged to 1 mA h cm$^{−2}$ at 0.2 mA cm$^{−2}$ to deposit Li to the negative electrode and subsequently discharged 2.5 V to strip the Li metal from the negative electrode, while the NMR spectra were continuously acquired. Single-pulse measurements with a π/2 pulse of 6.5 µs and recycle delay of 1 s was applied to acquire the 1D static spectrums. Each spectrum took ~2 min to acquire. The chemical shifts are referenced to a 0.1 M LiCl solution. Bruker Topspin 4.0.6 as well as Mestrenova were used for raw data processing and analysis.

**Electrical field calculations**. All simulations were done in COMSOL Multiphysics 5.4. Using the electrostatics software in the simulation a static, simplified, two-dimensional model of a battery is built. It shows the effects on the electric field around a dendrite when surrounded with different materials. The electrodes used were copper and lithium which were taken from the COMSOL material's library. Three different electrodes are considered, bare Cu and Cu with different insulator blocks (BTO, AO), with the top electrode having an electric potential of 1 V and the bottom electrode 0 V, thus acting as ground. Two electrodes (Cu (2 × 250 µm) versus BTO on Cu, or AO on Cu (2 × 250 µm)) were placed at two sides of the electrolyte (EC/DMC, 50 × 250 µm). The BTO and AO blocks on the electrode have the dimensions 8 × 8 µm with a gap of 2 µm. For the simulations, a dendrite is represented as a rectangular shape (0.1 × 0.2 µm) with a hemispherical top. The mesh uses triangular elements and is automatically generated by the program. The mesh size varies between 0.005 µm and 2.5 µm. The dielectric constants or equivalent the relative permittivity of the BTO, AO and the electrolyte EC/DMC is 4000, 8 and 40, respectively.

## Data availability
The data that support the findings within this paper are available from the corresponding author on request.

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

## Acknowledgements

The authors thank Michel Steenvoorden for assistance with the experiments. C.W. would like to thank the Guangzhou Elite Project for financially supporting in this paper. Financial support from the Advanced Dutch Energy Materials (ADEM) program of the Dutch Ministry of Economic Affairs, Agriculture and Innovation is gratefully acknowledged. Financial support is greatly acknowledged from the Netherlands Organization for Scientific Research (NWO) under the VICI grant nr. 16122.

## Author contributions

M.W. conceived the research, M.W. and C.W. designed the experiments. C.W. carried out materials and electrochemical characterization. M.L. and S.G. conducted NMR experiments and performed the data analysis. M.T. performed the electric field simulations. F.O. participated in part of the electrochemical characterization. M.W., C.W. and M.L. co-wrote the paper. C.W. and M.L. contributed equally to this work. All authors discussed the results and commented on the manuscript.

## Competing interests

The authors declare no competing interests.
