## [Peer Review File · Nature Communications]

REVIEWER COMMENTS

Reviewer #1 (Remarks to the Author):

The manuscript reports the use of BTO as a 3d host for lithium deposition to mitigate dendrite growth. BTO is a well known high dielectric constant material. Theoretically, it should change current distribution near the Li growth tip and generate more uniform coating. The idea is interesting and some of the claimed efficiency data are impressive. I have major reservations about the validity of some of the data and claims. and don't recommend its acceptance.

- 1) It is unclear to me how the 99.82% was calculated or measured. The plots in figure 3, with a scale of 0-100%, provide no support for the efficiency claims. This value is actually indeed shocking, since if it is true, it will represent a major breakthrough in the field.
- 2) The morphology shown in Figure 5 does not provide support for the claim of dense deposition and high efficiency either. The image in b shows a dark optical picture, a classic sign of mossy lithium. It is inconceivable that this morphology will lead to high efficiency;
- 3) If BTO indeed reacts with Li, what does it produce? How would the new phases impact the basic assumption of the paper, i.e., a modulation of electric field;
- 4) If the efficiency data are true, the full cell performance will be a major advancement for the field. Why are those not shown in the main text?

Reviewer #2 (Remarks to the Author):

I very much enjoyed reading this paper. The idea of diverting electric fields away from the dendrite tips is a very interesting one, in principle. Below you will find a few questions that as a reader I would find as loose ends that would detract from the great experimental science you have made. I strongly suggest to address them to help prop up the conceptual argument supporting the results.

The simulations seemed a bit simplistic to truly capture the effect of the surrounding environment and the dendrite tip on its shielding and growth. For example, it was not clear what were the applied boundary conditions in the presented simulations, what the mesh size is, long it took to complete, etc. This is necessary so that it can be reproduced, as one expect it should happen by future generations. It would be great if the coauthors added some information on the supplemental information. As presented, the computed fields seem unphysical large. In a real dendrite, the electric field in the vicinity of the dendrite tip is a result on the local overpotential, as induced by the local surrounding environment, the local electrochemical fields and the geometry of the dendrite itself, thus a field can be self-induced by a dendrite branch attempting to minimize its free energy.

Fundamentally, barium titanate, BTO, happens to be piezoelectric, so the application of electric fields (either through a counter electrode or as a result of an electrochemical potential gradient from a neighboring electrochemically active feature) could induce a strain (the converse piezoelectric effect) that would squeeze the dendrites (their yield point is very low), thus favoring dendrite growth. In addition, any stress that the BTO scaffold is subjected to, could induce a local electric field (direct piezoelectric effect) that could in turn induce an overpotential that would make a dendrite locally grow. Can the authors comment how on this and how it was averted? Based on the above, $D \neq \epsilon E$. Instead, $D = \epsilon E + d \sigma$, and $\sigma = C(S - dE)$, and the mechanical equilibrium equation would have to be solved too.

Thermodynamically, the high dielectric constant in the surrounding environment does not seem to be a condition strong enough to suppress the formation of a lithium (dendrite) phase. Not only it would require a small overpotential (as surpassed by surrounding BOT scaffold), but in general, a uniform electrochemical potential difference with the surrounding local environment. Thus a simple Gauss law

calculation might not be sufficient theoretical proof that the BTO scaffold can fully suppress the driving forces for dendrite nucleation. Please address this issue.

From the point of view of a device fabrication, the authors should comment how the foreign BTO scaffold would impact the device performance. In general, the addition of non-active material surpasses the energy density and increases the tortuosity of the electrode. How would this new and very exciting strategy, I should say, scale up to a real device. What does the scaffolding sacrifice in performance, and is it worth it?

Finally, if BTO scaffolding suppresses the dendrite formation through electric shielding, what is stopping any lithium that did accumulate between the BTO gaps to stay there, unused? It seems that it would lead to a new form of dead lithium. Please comment.

Experimentally, I was very excited to see the very impressive increased reversibility during galvanostatic cycling. Clearly, BTO scaffolding is working, but the presented argument needs to be tightened up so that it is not negatively criticized by the community.

Conceptually, it seems a contradiction that the BTO scaffold diverts (concentrates) the E-fields away from the dendrites, yet lithium occurs uniformly. These two facts need to be reconciled in the manuscript. Figure 6 does not help clear the question.

In the conclusion section, it is wrong to conclude that ferroelectricity (the reversible switching of polarization domains in BTO) has a dominant role in this process. Not only the simulations do not support any ferroelectric process, only shows simple dielectric processes, and thus should be rewritten.

There were some small English grammatical errors. Please review.

Overall, it is a great result, but the theoretical foundation that attempts to support it needs a full revision.

Reviewer #3 (Remarks to the Author):

Review of BTO scaffold Li Nat Comm

This manuscript describes a unique approach to suppress lithium dendrite growth by using a high dielectric constant layer deposited on the negative electrode (current collector). Encouraging results in lithium plating experiments that are attributed to the presence of the porous BTO layer make this work of significant interest to the lithium battery community. There are however some issues that should be addressed prior to further consideration of publication.

- An energy density of about 2000 mAh/g is estimated from the quantity of Li inserted into the porous BTO. This is misleading if intended to suggest the energy density of the whole battery, which would actually be dictated by the cathode. It should be explicitly stated that this number should be compared to the theoretical energy density of metallic Li.

- It is not clear why the authors need to invoke ferroelectricity – is it not the dielectric property that is responsible for influencing the electric field near the surface?
- Did the authors measure the dielectric constant of their BTO film, or did they use the bulk literature value?
- What specific method(s) was used to simulate the electric field? This can be put in SI section.
- More detail should be provided on the operando NMR set-up and experiment, rather than just a literature reference from another group. This is not a standard NMR method. This could also be relegated to SI.
- Was there any difference in SEI characteristics between the bare and BTO electrode, as determined by NMR? This would require examining the region near 0 ppm.
- It is stated that the cycling of the anodeless scaffold still eventually leads to short-circuit. More detail is needed; i.e., how many cycles and where and in which figure does it appear?
- There is some repetition at the end of Figure 2 caption.

Reviewer #1 (Remarks to the Author):

The manuscript reports the use of BTO as a 3d host for lithium deposition to mitigate dendrite growth. BTO is a well-known high dielectric constant material. Theoretically, it should change current distribution near the Li growth tip and generate more uniform coating. The idea is interesting and some of the claimed efficiency data are impressive. I have major reservations about the validity of some of the data and claims. and don't recommend its acceptance.

1) It is unclear to me how the 99.82% was calculated or measured. The plots in figure 3, with a scale of 0-100%, provide no support for the efficiency claims. This value is actually indeed shocking, since if it is true, it will represent a major breakthrough in the field.

Reply: We thank the reviewer for this important comment. We agree that these observations on the Coulombic Efficiency (CE) are not sufficiently addressed in the manuscript, and have completely revised this part of the manuscript, also adding additional data. The CE reported is the *average* value over all the tested cycles (including the initial cycles). We repeated these anodeless half cells tests during the period provided for the revision, which gave the same result, confirming the validity of our data. We do agree that we report an extra-ordinary (suspiciously) high efficiency, especially considering the LiPF₆ EC/DMC electrolyte. In fact, the anodeless BTO half-cells result in a 100% CE over hundreds of cycles, where one should realize that only the average Coulombic efficiency is meaningful as discussed below. For these results, the cycling protocol terminates Li metal stripping when the plated capacity is reached or if the voltage cut-off is reached. As can be observed in **Figure 3a**, the initial two cycles (having a very low coulombic efficiency of around 60 and 85%). Typically, for an anode less half-cell geometry the plated capacity is not reached during stripping, due to formation of dead Li metal and SEI products. This applies to the anodeless Cu and AO scaffold, and also for the first two cycles of the BTO scaffold. However, for the subsequent cycles the stripping of the BTO scaffold terminates because the plated capacity is reached (capacity limited cycling). This is the origin of the 100% coulombic efficiency after the first two cycles for BTO. In other words, the cycling appears as a

symmetric Li-metal cell that is cycled by limiting the capacity, although at the start the BTO scaffold did not contain Li metal. The 100% CE can thus be considered a consequence of the cycling protocol, but it only occurs for the BTO indicating it also reflects a significant difference in Li metal plating (as compared to bare Cu and the AO scaffold). We propose the following mechanism: a considerable amount of dead Li-metal forms during the first two cycles, which is reactivated (reconnected) during subsequent cycling, explaining why the stripping capacity is achieved before the voltage raises towards the Voltage cut-off (set at 1V vs Li/Li⁺). Thus a reserve of dead Li metal is formed during the first initial cycles, which is reactivated upon subsequent cycling. As a consequence the CE per cycle is not meaningful as the system turns into a symmetric Li metal geometry. What is meaningful, is that this reserve of Li metal build up during the first two cycles can be used for hundreds of cycles for BTO, indicating it can easily be reconnected to (consistent with more compact Li metal plating) and that less SEI is formed, both consistent with the operando solid state NMR observations. This better Li metal efficiency is captured by the average CE over all cycles, including the first two.

To further assess this we have also performed anode less half-cell cycling where the stripping always terminates by the voltage cutoff (at 1 V vs Li/Li⁺). This shows alternating CE's below and above 100% for BTO, which can also be explained by activation of dead Li metal that is build up in previous cycles.

Based on this we have completely rewritten the electrochemical cycling part of the manuscript, clearly defining the cycling protocol and how we report the CE. Cycling data of the half cells are provided for both cycling protocols, focussing on the impact of the protocol on the cycling, and describing the mechanism proposed above. As mentioned, all tests have been repeated during the revision time, more than once, providing consistently the same results.

2) The morphology shown in Figure 5 does not provide support for the claim of dense deposition and high efficiency either. The image in b shows a dark optical picture, a classic sign of mossy lithium. It is inconceivable that this morphology will lead to high efficiency;

Reply: We agree that the top SEM images do not provide support for dense deposition. In our view the operando NMR, shown in **Figure 2**, provides the most direct evidence for more compact plating in the presence of BTO. Additionally we show cross-section SEM images (**Figure 5g**), more convincingly indicating more dense deposition for the BTO samples as compared to the bare Cu substrate. The conclusions in this part of the manuscript are weakened, and mostly focussed on the cross section images.

3) If BTO indeed reacts with Li, what does it produce? How would the new phases impact the basic assumption of the paper, i.e., a modulation of electric field;

Reply: We thank the reviewer for this relevant question. We anticipate that a small part of the BTO is lithiated ($\text{Li}_x\text{BaTiO}_3$) thus leading to Ti^{3+} , which can be expected to lower the polarization and thus lower the dielectric constant, reducing the modulation of the electric field. Because its poor electronic conductivity we anticipate lithiation may only occur at the surface at most, as no change is observed in the bulk structure of the BTO after cycling with XRD (see **Figure S13**), and thus we anticipate the modulation of the electrical field is not significantly altered. We have added this to the discussion part of the revised manuscript.

4) If the efficiency data are true, the full cell performance will be a major advancement for the field. Why are those not shown in the main text?

Reply: This is a useful suggestion. We added the full cell data last minute to the submission, hence it ended up in the SI. In the complete revision of the part discussing the electrochemical cycling data, we have also moved part of the full cell cycling data to the main manuscript (in **Figure 3** and **Figure 4**) providing a much more comprehensive evaluation of the impact of the BTO scaffold.

Reviewer #2 (Remarks to the Author):

I very much enjoyed reading this paper. The idea of diverting electric fields away from the dendrite tips is a very interesting one, in principle. Below you will find a few questions that

as a reader I would find as loose ends that would detract from the great experimental science you have made. I strongly suggest addressing them to help prop up the conceptual argument supporting the results.

We would like to thank the reviewer for the positive and constructive comments regarding the conceptual arguments, which we address one by one below.

1. The simulations seemed a bit simplistic to truly capture the effect of the surrounding environment and the dendrite tip on its shielding and growth. For example, it was not clear what were the applied boundary conditions in the presented simulations, what the mesh size is, long it took to complete, etc. This is necessary so that it can be reproduced, as one expect it should happen by future generations. It would be great if the co-authors added some information on the supplemental information.

Reply: We fully agree that the simulations are too simplistic to capture the effect of the complex environment, especially locally at the double layer of the Li metal. Here the aim was to provide an indication of the impact of low and high dielectric materials on the electrical field lines in a porous scaffold, at the dendrite tip located micrometres away from the high dielectric volume. The details of the simulations have been added to the supplementary, including mesh size and boundary conditions. In the simulation a static, simplified, two-dimensional model of a battery is built. It shows the effects on the electric field around a dendrite when surrounded by materials having differences in dielectric constants. We agree that this does not provide a dynamic model where the actual growth of a dendrite is being investigated. The model consists of a top and bottom electrode, with the top electrode having an electric potential of 1 V and the bottom electrode 0 V, thus acting as ground. Between the electrodes is a 50 μm electrolyte layer. The average electric field is therefore 2×10^4 V/m ($1 \text{ V} / 50 \mu\text{m}$); around the modelled dendrite it can be five times larger depending on its surrounding. The width of the battery is 250 μm ; the high/low dielectric volumes measure $8 \times 8 \mu\text{m}^2$; the dendrite measures $0.1 \times 0.2 \mu\text{m}^2$. Therefore the battery can be assumed to be infinitely wide from the point of view of the dendrite and dielectric volumes. The mesh uses triangular elements and is automatically generated by the program. The mesh size varies between 0.005 μm and 2.5 μm . Since it is a static, 2D model it takes less than a minute the solve.

2. As presented, the computed fields seem unphysical large. In a real dendrite, the electric field in the vicinity of the dendrite tip is a result on the local overpotential, as induced by the local surrounding environment, the local electrochemical fields and the geometry of the dendrite itself, thus a field can be self-induced by a dendrite branch attempting to minimize its free energy.

Reply: We agree these simulations are greatly simplified as compared to the real situation, which is a complex mix of overpotentials and the electrochemical double layer. Therefore we wish to stress that we do not claim that the electrical field calculated at the tip of the dendrites is fully realistic providing a quantitative measure. Also, we agree that these simple simulations based on Gauss law, are a proof that a high dielectric constant can suppress dendrites. They merely serve as an indication that in the vicinity of a high dielectric volume, the electrical field can be expected to be reduced. If this in practice dominates over the local effects at the surface of the dendrites is indeed a relevant question. But if one assumes that larger electrical field at the tip of a dendrite plays a role as driving force for dendrite growth, and that a high dielectric in the vicinity can be expected to lower the electrical field gradient, we arrive at the hypothesis of the manuscript. The experimental results, especially the more dense plating as observed by operando NMR in **Figure 2** appear to support this, which is in our view the most important result of our manuscript. To make this more clear we have stressed to simplified character, limitations of the simulations in the manuscript when discussing the simulation results.

3. Fundamentally, barium titanate, BTO, happens to be piezoelectric, so the application of electric fields (either through a counter electrode or as a result of an electrochemical potential gradient from a neighboring electrochemically active feature) could induce a strain (the converse piezoelectric effect) that would squeeze the dendrites (their yield point is very low), thus favoring dendrite growth. In addition, any stress that the BTO scaffold is subjected to, could induce a local electric field (direct piezoelectric effect) that could in turn induce an overpotential that would make a dendrite locally grow. Can the authors comment how on this and how it was averted? Based on the above, $D \neq \epsilon E$. Instead, $D = \epsilon E + d \sigma$, and $\sigma = C(S - dE)$, and the mechanical equilibrium equation would have to be solved too.

Reply: We thank the reviewer for addressing this important issue. The model is static. And it looks only at the electric fields for a given, assumed and simplified situation. It doesn't take in to consideration any mechanical, chemical or piezo-electric effects. It is used to support the hypothesis that the BTO layer lowers the field gradient around the dendrite. It is not used to directly show that the growth is limited by the BTO layer. That is a conclusion made separately from the simulation. Regarding the piezoelectricity, we thank the reviewer for bringing forward this important consideration. Although the piezoelectric coefficients are relatively large for BTO, the electrical fields within the battery are very small. Considering that the electrical fields at the BTO do not exceed 10^5V/m in the simulations, which likely overestimate the electrical field, the strain can be expected well below 1%.¹ Since BTO is casted into a very porous layer (>60%) through a binder, we propose that this does not give rise to significant compression of the dendrites. In other words we think it is reasonable to assume $D = \epsilon E$. We have addressed this reasoning in the revised version of the manuscript.

4. Thermodynamically, the high dielectric constant in the surrounding environment does not seem to be a condition strong enough to suppress the formation of a lithium (dendrite) phase. Not only it would require a small overpotential (as surpassed by surrounding BOT scaffold), but in general, a uniform electrochemical potential difference with the surrounding local environment. Thus a simple Gauss law calculation might not be sufficient theoretical proof that the BTO scaffold can fully suppress the driving forces for dendrite nucleation. Please address this issue.

Reply: This relates to comment 2. We agree that our simple Gauss law calculation does not prove dendrite suppression. As mentioned in above reply's, the purpose of the calculations is to indicate that in the vicinity (micrometer range) of a high dielectric the electrical field gradients decrease near Li-metal inhomogeneity's. In combination with the observed more compact Li-metal plating, and more efficient electrode cycling, we argue that the high dielectric constant of the BTO lowers dendrite formation, where the lowered electrical field gradient provides a consistent rational. Especially important in our view is the comparison with a similar porous Al_2O_3 scaffold, having a low dielectric constant, which does not result in the more compact plating and improved performance. We have revised the discussion of

the calculations to reflect these considerations, stressing the limitations of the simple simulations.

5. From the point of view of a device fabrication, the authors should comment how the foreign BTO scaffold would impact the device performance. In general, the addition of non-active material surpasses the energy density and increases the tortuosity of the electrode. How would this new and very exciting strategy, I should say, scale up to a real device. What does the scaffolding sacrifice in performance, and is it worth it?

Reply: Indeed the BTO scaffold sacrifices specific capacity by the added weight. For the investigated electrodes the BTO scaffold has a porosity of 60%, this lowers the specific electrode capacity from 3860 mA h g⁻¹ (pure Li-metal) to 2000 mA h g⁻¹ (pure Li metal + weight of the BTO). We deliberately produced a relatively porous scaffold (~60%), to promote a high practical specific capacity. Possibly the porosity can be lowered even further to raise the specific capacity further, which is just one of the parameters that needs further investigation to investigate practical feasibility of this strategy.

6. Finally, if BTO scaffolding suppresses the dendrite formation through electric shielding, what is stopping any lithium that did accumulate between the BTO gaps to stay there, unused? It seems that it would lead to a new form of dead lithium. Please comment.

Reply: This is indeed a relevant comment. Dendrites are likely to lead to “dead” lithium because stripping leads to disconnected fragments of Li metal.^{2,3} The more compact the Li-metal plating is, the less likely it will lose contact to the current collector (the stripping is forced to be more top-down when the Li-metal is less porous). Because the plating within the 60% porous volume of the BTO appears to be very compact, we argue this lowers the formation of dead Li-metal, as directly observed by the operando NMR shown in **Figure 2**. Also the revised text on the anodeless half-cell cycling (see reply to the comment of Reviewer #1), explicitly discusses the role of dead Li metal in BTO, where it is concluded that in BTO dead Li metal can more easily be reactivated, presumably the result of the more compact nature of the Li metal deposits in the BTO.

7. Experimentally, I was very excited to see the very impressive increased reversibility during galvanostatic cycling. Clearly, BTO scaffolding is working, but the presented argument needs to be tightened up so that it is not negatively criticized by the community.

Reply: We thank the reviewer for the constructive comment. Using the input from the reviewers we have attempted to be more concise and careful in discussion of the cycling data of the half-cells. As indicated at comment 1 of reviewer #1, this section is completely rewritten, and additional cycling data is provided, that enables a much more detailed discussion based on the observations, including the impact of the cycling protocol, and the role of dead Li metal, and the impact of the BTO scaffold on this.

8. Conceptually, it seems a contradiction that the BTO scaffold diverts (concentrates) the E-fields away from the dendrites, yet lithium occurs uniformly. These two facts need to be reconciled in the manuscript. Figure 6 does not help clear the question.

Reply: We are not sure how this poses a contradiction. The high dielectric phase displaces the electrical field gradient away from the pores within the scaffold, thus promoting more compact plating within these pores (as visualized in **Figure 1**).

9. In the conclusion section, it is wrong to conclude that ferroelectricity (the reversible switching of polarization domains in BTO) has a dominant role in this process. Not only the simulations do not support any ferroelectric process, only shows simple dielectric processes, and thus should be rewritten.

Reply: We thank the reviewer for indicating the erroneous association of dielectric with ferroelectricity (which typically go hand in hand), where indeed the dielectric process is considered in the manuscript. This is corrected.

10. There were some small English grammatical errors. Please review.

Reply: We have checked and revised the manuscript carefully.

Reviewer #3 (Remarks to the Author):

This manuscript describes a unique approach to suppress lithium dendrite growth by using a high dielectric constant layer deposited on the negative electrode (current collector).

Encouraging results in lithium plating experiments that are attributed to the presence of the porous BTO layer make this work of significant interest to the lithium battery community.

There are however some issues that should be addressed prior to further consideration of publication.

1. An energy density of about 2000 mAh/g is estimated from the quantity of Li inserted into the porous BTO. This is misleading if intended to suggest the energy density of the whole battery, which would actually be dictated by the cathode. It should be explicitly stated that this number should be compared to the theoretical energy density of metallic Li.

Reply: We agree. The 2000 mAh/g represents the specific capacity of the anode, and only in combination with the specific capacity and potential of the cathode an energy density can be estimated.

2. It is not clear why the authors need to invoke ferroelectricity – is it not the dielectric property that is responsible for influencing the electric field near the surface?

Reply: We thank the reviewer for indicating the erroneous mentioning of ferroelectricity, where dielectric was intended. This is corrected.

3. Did the authors measure the dielectric constant of their BTO film, or did they use the bulk literature value?

Reply: The dielectric constant was provided by the producer of the BTO material (Euro Support B.V.).

4. What specific method(s) was used to simulate the electric field? This can be put in SI section.

Reply: We agree not sufficient details of the simulations were provided. In the simulation a static, simplified, two-dimensional model of a battery is built. The model consists of a top and bottom electrode, with the top electrode having an electric potential of 1 V and the bottom electrode 0 V, thus acting as ground. In between the electrode is a 50 μm electrolyte layer. The average electric field is therefore $2 \cdot 10^4$ V/m (1 V / 50 μm); around the modelled dendrite it can be five times larger depending on its surrounding. The width of the battery is 250 μm ;

the shielding blocks measure $8 \times 8 \mu\text{m}^2$; the dendrite measures $0.1 \times 0.2 \mu\text{m}^2$. Therefore the battery can be assumed to be infinitely wide from the point of view of the dendrite and shielding blocks. The mesh uses triangular elements and is automatically generated by the program. The mesh size varies between $0.005 \mu\text{m}$ and $2.5 \mu\text{m}$. We have added this information to the SI.

5. More detail should be provided on the operando NMR set-up and experiment, rather than just a literature reference from another group. This is not a standard NMR method. This could also be relegated to SI.

Reply: We thank the reviewer for this suggestion. We have provided the following details on the NMR experiment: LCO/Cu and LCO/BTOCu plastic capsule cells, (designed for operando NMR measurements) with a conventional carbonate electrolyte were assembled in the glove box and studied by operando ^7Li -NMR to monitor the microstructural evolution of Li deposits. Measurements were done on a wide bore Bruker Ascend 500 system equipped with a NEO console with a magnetic field strength of 11.7T and a ^7Li resonance frequency of 194.37 MHz. Operando static ^7Li NMR experiments were performed at room temperature with an NMR Service ATMC operando NMR probe, and the electrochemical cell was simultaneously controlled by a portable Maccor battery testing system. During the 1D static ^7Li NMR measurements the cells were charged to 1 mA cm^{-2} at 0.2 mA cm^{-2} to deposit Li to the anode and subsequently discharged 2.5 V to stripped the Li-metal from the anode, while the NMR spectra were continuously acquired. Single-pulse with a $\pi/2$ pulse of $6.5 \mu\text{s}$ and recycle delay of 1 s was applied to acquire the 1D static spectrums. Each spectrum took ~ 2 minutes to acquire. The chemical shifts are referenced to a 0.1M LiCl solution. Bruker Topspin 4.0.6 as well as Mestrenova were used for raw data processing and analysis. These details have been added to the methods section

6. Was there any difference in SEI characteristics between the bare and BTO electrode, as determined by NMR? This would require examining the region near 0 ppm.

Reply: We thank the reviewer for this valuable suggestion. We compared the 1D NMR spectra of the in situ NMR experiments around 0 ppm chemical shift, as shown in Figure below. It is difficult to distinguish the SEI species from the electrolyte resonance. However, as can be observed from the peak intensity change before and after 3 cycles, there appears more Li salt

consumption in the LCO-Cu battery compared to that in the LCO-BTO-Cu battery, consistent with less severe Li dendrite growth in the LCO-BTOCu battery. Above discussion has been added to the revised manuscript, and the data shown below is provided as **Figure S8** in the supporting information.

Figure S8 Anode-less (a) Cu and (b) BTO scaffold on Cu versus a LiCoO₂ cathode with a 1M LiPF₆ EC/DMC electrolyte cycled at 0.2 mA cm⁻² to 1 mA cm⁻² charge capacity and discharge to 2.5 V cut-off 1D ⁷Li solid-state NMR spectra at selected conditions (50 to -50 ppm).

7. It is stated that the cycling of the anode less scaffold still eventually leads to short-circuit. More detail is needed; i.e., how many cycles and where and in which figure does it appear?

Reply: We thank the reviewer for this useful suggestion to provide more details. As shown in **Figure 4a** and **Figure S10**, cells with BTO scaffold cycling at 2 mA cm⁻² and 4 mA cm⁻² were short-circuited at cycle 241th and at cycle 80th respectively, while At 8 mA cm⁻² the cell fails due to the high polarization at cycle 30th. At the higher cycling capacity, 4mAh cm⁻², and at current density the short circuited at cycle 167th. Above details has been added to the revised manuscript.

8. There is some repetition at the end of Figure 2 caption.

Reply: We thank the reviewer for spotting this. The repetition in the caption of figure has been removed.

Reference:

- 1 Gao, J., Xue, D., Liu, W., Zhou, C. & Ren, X. in *Actuators*. 24 (Multidisciplinary Digital Publishing Institute).
- 2 Lv, S. *et al.* Operando monitoring the lithium spatial distribution of lithium metal anodes. *Nature communications* **9**, 1-12 (2018).
- 3 Tewari, D., Rangarajan, S. P., Balbuena, P. B., Barsukov, Y. & Mukherjee, P. P. Mesoscale anatomy of dead lithium formation. *The Journal of Physical Chemistry C* **124**, 6502-6511 (2020).

REVIEWER COMMENTS

Reviewer #1 (Remarks to the Author):

I appreciate the effort made by the authors in addressing the comments. However, my assessment of the manuscript did not change. While the concept of using BTO is interesting, the electrochemical data are of very questionable quality that I would not recommend publication.

1. The charge-discharge profiles in Figure S9 need to have the corresponding current trace. If it was a constant current square wave, the voltage response of the BTO made no sense. The curves from other substrates are also barely readable. Moreover, Figure 3b shows the highly unusual fluctuating efficiency data. What are the charge-discharge profiles for those specific cycles? I suspect there was severe shorting, which led to the artificially high coulombic efficiency.

2. Figure 3d full cell data are obviously inconsistent with half cell data. The large amount of trapped lithium during prelithiation also makes assessing the actual efficiency difficult. If dead lithium can indeed be reconnected, why does that not happen in the full cell?

3. BTO has a density of $\sim 6 \text{ g/cm}^3$. At 60% porosity, how was the 2000 mAh/g capacity calculated?

Reviewer #3 (Remarks to the Author):

The authors have addressed my concerns adequately, except that they should add a very brief note to the text about the 2000 mAh/g in addition to simply agreeing with me in the rebuttal. If the other reviewers are satisfied with the responses and revisions, I am supportive of acceptance and do not need to see the manuscript again.

RESPONSE TO REVIEWERS

Reviewer #1 (Remarks to the Author):

I appreciate the effort made by the authors in addressing the comments. However, my assessment of the manuscript did not change. While the concept of using BTO is interesting, the electrochemical data are of very questionable quality that I would not recommend publication.

1. The charge-discharge profiles in Figure S9 need to have the corresponding current trace. If it was a constant current square wave, the voltage response of the BTO made no sense. The curves from other substrates are also barely readable. Moreover, Figure 3b shows the highly unusual fluctuating efficiency data. What are the charge-discharge profiles for those specific cycles? I suspect there was severe shorting, which led to the artificially high coulombic efficiency.

Reply:

We agree with the reviewer that Figure S9 should be made clearer. We have increased the figures for clarity. Adding the current trace will make it even more complicated because of the different timelines of the three substrates. What is important to realize is that there was a rest of 2 to 30 minutes between each period of constant current (as described in the methods section). To help the reader, we now indicated parts of the voltage curve where the rest (zero current) takes place, and indicated this also in the caption of Figure S9.

To support the origin of the fluctuation of the efficiency in Figure 3b, we added the voltage limited stripping data (belonging to the date in Figure 3b) as **Figure S15**. The smooth voltage profiles demonstrate that these cells do not short. This conclusion, and reference to **Figure S15** is added to the main manuscript on page 15.

In our view the best explanation of the fluctuating efficiency is the build-up of the Li metal reserve during the first cycles, which is reconnected during the subsequent cycles, as set out in detail in the revised manuscript (page 15).

2. Figure 3d full cell data are obviously inconsistent with half-cell data. The large amount of trapped lithium during prelithiation also makes assessing the actual efficiency difficult. If dead lithium can indeed be reconnected, why does that not happen in the full cell?

Reply: This is an important comment. The situation is very different in the full and half cells, not only because of the presence of the cathode which can cause cross over of transition metal ions from the NMC811 cathode, and provide other sources of irreversible capacity loss. Also, after a low coulombic efficiency cycle, leaving dead Li, in the half cell situation again the full capacity is plated (because of the practically infinite capacity of the counter Li-metal electrode), which is not available in the full cell. Thereby the reconnecting effect can be expected to be smaller in full cells. In Figure 1(a) below we show further cycling where at a number of cycles, indicated by the arrows, have a larger than 100% CE. Also in this case the voltage profiles look normal, indicating no short circuiting. We suggest that this represents the reconnecting phenomena in the full cells.

Figure 1 (a) Cycling performance of Li-metal and BTO scaffold when paired with NCM cathode and (b) voltage profiles of which have the CE over 100%.

3. BTO has a density of $\sim 6 \text{ g/cm}^3$. At 60% porosity, how was the 2000 mAh/g capacity calculated?

Reply: We thank the reviewer for this comment. Here we made a calculational fault. The specific capacity is $(2046/(0.53+((1/p)-1)))$ using the density of BTO (6.02 g/cm^3), Li metal (0.53 g/cm^3) and the specific capacity of Li metal (3860 mAh/g) and where p represents the fractional porosity (between 0 and 1). Additionally, the manuscript reported the porosities of early results, and not of the latest results where the electrochemistry is shown for. These porosities are 74%. All together this results in a specific capacity of approximately 800 mAh/g taking into account the weight of the BTO, and shows that reaching 2000 mAh/g requires a porosity of 0.92. These results are now corrected and updated throughout the manuscript.

Reviewer #3 (Remarks to the Author):

The authors have addressed my concerns adequately, except that they should

add a very brief note to the text about the 2000 mAh/g in addition to simply agreeing with me in the rebuttal. If the other reviewers are satisfied with the responses and revisions, I am supportive of acceptance and do not need to see the manuscript again.

Reply: We are sorry for this oversight. The capacity calculation is now added in the methods section.